



# NH₃ Converts Criegee Intermediates to Nitrogenous Organics

Xiaoying Li[1,2], Long Jia[1,2], Yongfu Xu[2,3]

[1]State Key Laboratory of Atmospheric Environment and Extreme Meteorology, Institute of Atmospheric Physics, Chinese Academy of Sciences, Beijing, 100029, China

[2]College of Earth and Planetary Sciences, University of Chinese Academy of Sciences, Beijing, 100049, China

[3]State Key Laboratory of Atmospheric Boundary Layer Physics and Atmospheric Chemistry, Institute of Atmospheric Physics, Chinese Academy of Sciences, Beijing, 100029, China

*Correspondence to*: Long Jia (jialong@mail.iap.ac.cn)

**Abstract graphic**

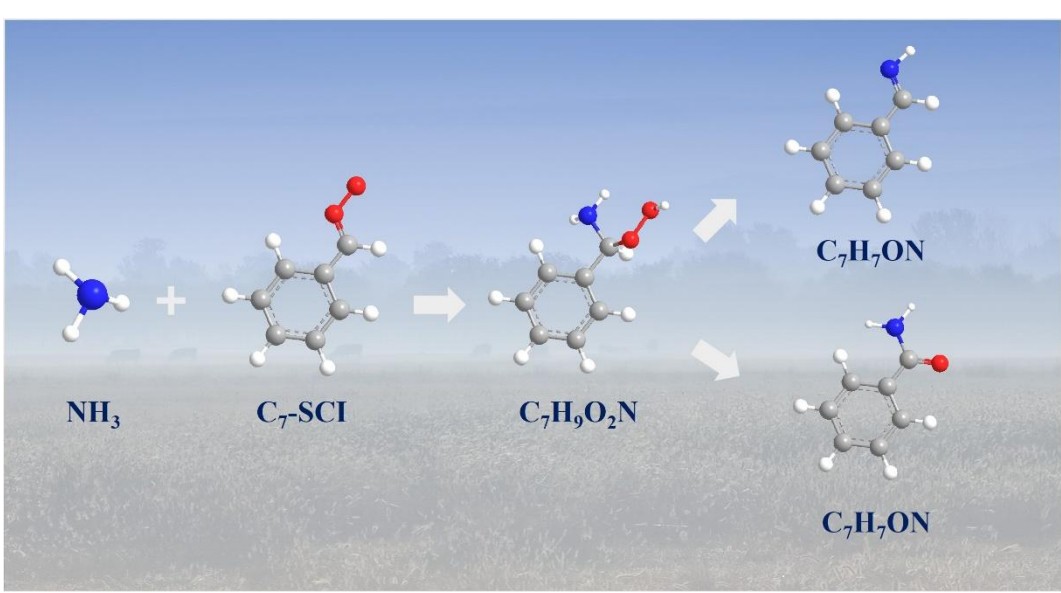

**Abstract.** Ammonia (NH₃), the dominant alkaline gas in the atmosphere, plays a critical role in urban air quality, but its molecular-level interactions with organics remain poorly understood. Here, we uncover a hidden chemical pathway: NH₃ efficiently scavenges stable Criegee intermediates (SCI) - critical radical in organic aerosol formation. Using high-resolution

Orbitrap mass spectrometry, we capture the first real-time evidence of NH₃ reacting with styrene-derived C₇-SCI to form a hazardous peroxide amine ($C_7H_9O_2N$) while suppressing traditional SCI-driven aerosol components like benzoic acid and oligomers. Due to unstable bond of peroxide in the molecule, $C_7H_9O_2N$ can further decompose into more stable compounds (imine $C_7H_7N$ and amide $C_7H_7ON$). This study discovered a critical reaction pathway for the formation of organic amines through the reaction of NH₃ and SCI, which not only bridges a critical gap in understanding NH₃'s role in aerosol chemistry

but also exposes a previously overlooked health risk from nitrogen-enriched particulate matter.



## 1 Introduction

Secondary organic aerosols (SOA) are critical components of atmospheric fine particles, typically formed by the oxidation of volatile organic compounds (VOCs) (Ehn et al., 2014; Hallquist et al., 2009). SOA can significantly impact air quality and
climate by scattering and absorbing sunlight, and affect human health due to their ability to reach deep into lungs(Calvin et al., 2023; Kroll and Seinfeld, 2008). Among SOA components, nitrogen-containing organic compounds (NOCs) are of particular importance due to their potential toxicity and role in light absorption(Laskin et al., 2025; Li et al., 2025; Yu et al., 2024c). Ammonia ($NH_3$) is the most abundant alkaline gas in the atmosphere and plays a significant role in aerosol chemistry(Behera et al., 2013; Krupa, 2003). Global $NH_3$ emissions have been increasing in recent years, largely due to agricultural and industrial
activities, yet models have not accounted for its potential to influence SOA(Fu et al., 2017; Meng et al., 2020; Zhang et al., 2023). $NH_3$ is known to enhance SOA yields by acid-base reactions(Du et al., 2023; Li et al., 2018; Lv et al., 2022; Zhang et al., 2023), and previous studies have focused on NOCs formation via reactions between $NH_3$ and carbonyl compounds(Laskin et al., 2014; Liu et al., 2021, 2023). Our recent study has shown new evidence that $NH_3$ can also react with isoprene-derived stable Criegee intermediates (SCIs) to form NOCs, thereby changing the chemical characteristics of SOA(Li et al., 2024b).
Styrene is an important anthropogenic VOC emitted from industrial processes and vehicle exhaust(Cui et al., 2022; Okada et al., 2012), and is a key precursor to urban SOA(Sun et al., 2016; Wu and Xie, 2018). Styrene ozonolysis can generate two types of SCI, namely $C_1$-SCI ($CH_2OO$) and $C_7$-SCI ($C_7H_6OO$) (Tuazon et al., 1993). Our studies have shown that $C_1$ and $C_7$ - SCIs play a key role in SOA formation through oligomerization(Tajuelo et al., 2019; Yu et al., 2022). Styrene is a unique aromatic with both aromatics and alkenes properties due to the containing of an aromatic ring and a highly reactive double
bond in the molecule. Our recent study revealed that $NH_3$ can greatly suppress biogenic SOA formation from isoprene by the reaction with SCIs, which can change pathways from oligomerization to the formation of small molecular nitrogenous products(Li et al., 2024b). However, it is still unknown whether this mechanism is applicable to all alkenes, especially anthropogenic sources of aromatic hydrocarbons such as styrene.

In this study, we investigate the reactions between $NH_3$ and styrene-derived products and their role in SOA formation.
Combining chamber experiments, molecular-level measurements through Orbitrap-MS, and iodometry kinetic control experiments, we confirm that $NH_3$ can react with Criegee intermediates to form a peroxide amine ($C_7H_9O_2N$) and identify its decomposition products ($C_7H_7N$ and $C_7H_7ON$). Our results reveal a common pathway in both biogenic and anthropogenic alkene VOCs, where $NH_3$ can change Criegee intermediates chemistry toward nitrogen-containing products with reactive peroxide, which may enhance aerosol toxicity. This study bridges a critical gap in understanding the role of $NH_3$ in urban
aerosol chemistry and highlights the need to refine SOA predictions in $NH_3$-polluted regions.

## 2 Materials and methods

**Chamber experiments**: All the experiments were conducted in FEP reactors under dark conditions, with background air supplied by purified zero air. Here is only a brief introduction, and additional chamber details are provided in the supplementary





material (Li et al., 2024b; Yu et al., 2024a). Styrene was injected into the reactor with zero air using a glass microsyringe, $O_3$ was produced by an ozone generator with pure $O_2$, and $NH_3$ was directly injected into the reactor. Experiments 1-5 were conducted in a 1.2 $m^3$ chamber, monitoring the particle concentration and size, with particles collected on a 25 mm PTFE membrane and extracted with pure methanol. Experiments 6-10 were performed in a 150 L chamber to obtain the molecular composition changes of SOA by online measurements. Due to the selective reaction of $I^-$ ions with peroxide bonds, Experiment 11 was conducted in a 1.2 $m^3$ chamber for iodometry kinetic experiments. The particles were collected and extracted with acetonitrile, and then treated with KI to determine the peroxides (Li et al., 2024a). Detailed experimental conditions are provided in Table S1.

**Measuring instruments**: Styrene was measured by a proton transfer reaction-mass spectrometer (PTR-MS P1000-L-AI, Anhui Province Key Laboratory of Medical Physics and Technology), and $O_3$ was measured with an $O_3$ analyzer (Model 49C, Thermo Scientific). The particle concentrations and size distributions were determined by a scanning mobility particle sizer (Model 3936, TSI). The gas-phase organics were online ionized by a gas aerosol in-situ ionization source (GAIS), and then measured by a high-resolution Orbitrap mass spectrometer (Orbitrap MS, Q-Exactive, Thermo Scientific). The particle-phase organics were injected and separated by a high-performance liquid chromatography (HPLC, Thermo Scientific), ionized by a heated electrospray ionization (ESI) source, and then the molecular composition were measured by Orbitrap MS.

**Data Analysis and Toxicity calculation**: Raw spectra were processed using Xcalibur (v4.1.31.9, Thermo Scientific). Tandem MS (MS²) was used to determine molecular structures, and Mass Frontier (v7.0.5.9, Thermo Scientific) can simulate potential product ions for molecule with known structure, which were then compared to the $MS^2$ spectra of molecular ion species to confirm the final structures of the molecules. To evaluate the influence of $NH_3$-SCI reactions, $NH_3$-related reactions were added into the Master Chemical Mechanism and simulated(Bloss et al., 2005; Jenkin et al., 2003; Jia et al., 2023; Jia and Xu, 2021). The OECD QSAR Toolbox (Version 4.7, http://qsartoolbox.org) is used for the calculation of molecular toxicity, and additional details are presented in the supplementary material.

## 3 Results and discussion

### 3.1 $NH_3$ suppresses SOA formation from styrene

As $NH_3$ concentrations increased, SOA yields decreased significantly from 4.9% (0 ppm $NH_3$) to 1.0% (0.8 ppm $NH_3$), showing an obvious inhibitory effect (Fig.1a). The observed yields with 0 ppm $NH_3$ are within the range of those previously reported for styrene ozonolysis under no $NH_3$ conditions(Bracco et al., 2019; Díaz-de-Mera et al., 2017; Yu et al., 2022, 2024b), which demonstrates the rationality of our experiment. A strong negative correlation was observed between $NH_3$ levels and SOA yields ($R^2$=0.98), confirming significant suppression of SOA by the presence of $NH_3$. A linear decrease in SOA yields indicates that $NH_3$ directly breaks up the key pathways of SOA formation in styrene ozonolysis.

MS analysis reveals that $NH_3$ suppresses SOA formation by affecting oligomerization pathways of styrene-derived SCIs. As shown in Fig.1b, the most significant peaks $C_6H_{14}O_9Na^+$ (m/z=253.052), $C_5H_{12}O_7Na^+$ (m/z=207.047) and $C_{13}H_{20}O_{11}Na^+$





(m/z=375.089) exhibit regular mass differences corresponding to $C_1$-SCI ($CH_2OO$, $\Delta$ m/z = 46.005) and $C_7$-SCI ($C_7H_6OO$, $\Delta$ m/z=122.036), consistent with our previous works that the oligomerization of SCIs is the main mechanism for SOA formation from styrene(Yu et al., 2022). These oligomers are significantly reduced by 51% with increasing $NH_3$ concentrations, which strongly supports the result that $NH_3$ can inhibit the formation of oligomers from SCIs.

90  Meanwhile, benzoic acid ($C_7H_5O_2^-$, m/z=121.028), as the dominant compound in styrene-ozonolysis system, is significantly suppressed by 51% with increasing $NH_3$ concentration (Fig.1c), which is consistent with the trend of SOA yield inhibition (50%). Since benzoic acid is mainly formed from the reaction of $C_7$-SCI with $H_2O$, the presence of $NH_3$ apparently competes with $H_2O$ for SCIs and inhibits the formation of benzoic acid. In addition, to confirm the inhibition of $NH_3$ competing with $H_2O$ for $C_7$-SCI on benzoic acid, we further conducted comparative online observations. Experimental observation show that

95  benzoic acid was suppressed by over 90% under high $NH_3$ concentration (10 ppm) and low relative humidity (2%) conditions (Fig.1d). The simulation results from MCM show that high concentration $NH_3$ (10 ppm) can suppress benzoic acid formation over 70% at 2%RH, and 50% inhibition at 17%RH. This inhibition intensifies to 80% when comparing high-$NH_3$/2%RH to low-$NH_3$/17%RH conditions. The consistency between the results from ESI, GAIS and MCM simulation confirms the role of $NH_3$ in competitively reacting with $C_7$-SCI.

100  Since $C_7$-SCI-derived products (oligomers and benzoic acid) were greatly suppressed with the presence of $NH_3$, where did $C_7$-SCIs go? Our previous study on the isoprene-ozonolysis system found that $NH_3$ can react with SCIs to produce amines, thereby inhibiting the original oligomerization pathway of SCIs and reducing SOA yields(Li et al., 2024b). This is consistent with the phenomenon observed in this study and may be due to the same mechanisms, indicating that the reaction between $NH_3$ and SCIs may be common in alkenes.





**Figure 1: SOA yields from styrene ozonolysis under different NH₃ concentrations (a); Positive mode mass spectra of SOA from styrene ozonolysis systems with 0 ppm (blue) and 0.4 ppm NH₃ (red) (b), several top ion peaks assigned to SCI-derived oligomer are marked in black; The mass spectra of benzoic acid from styrene ozonolysis systems with 0 ppm (blue) and 0.4 ppm NH₃ (red) (c); Online observation of benzoic acid in the experiments with low concentration NH₃ with normal humidity (Ex.8, blue) and high concentration NH₃ with low humidity (Ex.10, red) (d).**



### 3.2 Validation of the reaction pathway between NH$_3$ and SCI

Referring to the reaction mechanism between C$_4$-SCI and NH$_3$ from isoprene,(Li et al., 2024b) C$_7$-SCI should react with NH$_3$ to produce a molecule C$_7$H$_9$O$_2$N. Online GAIS-Orbitrap MS measurements identified a nitrogen-containing product at m/z 140.071 with the molecular formula C$_7$H$_{10}$O$_2$N$^+$ (Fig.2a), which is in good agreement with the predicted product C$_7$H$_9$O$_2$N.

However, it should be noted that the ammonium adduct ion of benzoic acid is also 140, and its molecular formula is the same as C$_7$H$_{10}$O$_2$N$^+$ (C$_7$H$_6$O$_2$+NH$_4^+$, m/z 140.071). We worried that this might affect the determination of C$_7$H$_{10}$O$_2$N$^+$. Therefore, to rule out the potential interference introduced by benzoic acid-ammonium adducts, we first compared the MS$^2$ spectra of m/z 140.071 (C$_7$H$_{10}$O$_2$N$^+$) and benzoic acid. Since ammonium ions are easily separated, the MS$^2$ of the ammonium adduct ion of benzoic acid may be mainly from m/z 123.044 (C$_7$H$_7$O$_2^+$). Results show that the MS$^2$ C$_7$H$_{10}$O$_2$N$^+$ (m/z 140.071) is different to the MS$^2$ C$_7$H$_7$O$_2^+$ (m/z 123.044) (Fig.2b). Different MS$^2$ spectra confirm that the molecule C$_7$H$_{10}$O$_2$N$^+$ is a unique new amine species, rather than an ammonium adduct derived from benzoic acid. We also conducted online observations by introducing NH$_3$ into pure benzoic acid vapor and found that no signal at m/z 140.071was detected, thus excluding the possibility of adducts. These prove that (C$_7$H$_{10}$O$_2$N$^+$) is not an adduct ion of benzoic acid and NH$_4^+$, but a newly generated species.

Based on our previous study on the reaction mechanism between NH$_3$ and SCI from isoprene, the molecule C$_7$H$_{10}$O$_2$N$^+$ (m/z 140.071) should contain a peroxide bond. To determine the presence of peroxide bond in the molecule of C$_7$H$_{10}$O$_2$N$^+$, we further conducted iodometry kinetic experiments based on the selective reaction of I$^-$ ions with peroxide bonds. The chromatographic results of iodometry kinetic experiments showed that the peak of C$_7$H$_{10}$O$_2$N$^+$ appeared at 21.07 min. While in the control sample with added KI, its peak intensity at 21.07 min was suppressed by almost 100% (Fig.2a). This verifies the presence of a peroxide bond in C$_7$H$_{10}$O$_2$N$^+$, and meanwhile also confirms the molecular structure of the product from the reaction between C$_7$-SCI and NH$_3$.





**Figure 2: Time series of online observation of C₇H₁₀O₂N⁺(a), and the chromatograms of molecule C₇H₁₀O₂N⁺ (m/z 140.071) are shown as an inset with the initial non-KI-treated sample (black) and KI-treated sample (red). MS² spectra of m/z 140.071 (C₇H₁₀O₂N⁺, blue) and m/z 123.044 (C₇H₇O₂⁺, black) in positive modes (b). Time series of online observation of C₇H₈N⁺ and C₇H₈ON⁺ (c,e). The comparison of the ion peaks in the MS² spectra of C₇H₈N⁺ and C₇H₈ON⁺ (black bars) with the major simulated product ions of C₇H₈N⁺ and C₇H₈ON⁺ (red bars) (d,f). The mechanism of NH₃ effects on SOA from styrene ozonolysis in this study (g).**



### 3.3 Fate of the products of NH₃ and SCI

Since $C_7H_{10}O_2N^+$ contains a peroxide bond, which makes it unstable and likely to decompose to form more stable compounds.

Referring to the general decomposition principle of peroxides, the peroxide amine $C_7H_{10}O_2N^+$ should further decompose into imines and amides due to its strong activity. Online MS measurements detected an imine $C_7H_8N^+$ (m/z=106.066) and an amide $C_7H_8ON^+$ (m/z=122.060) as the dominant products (Fig.2 c, e). We compared the MS² spectra of $C_7H_8N^+$ and $C_7H_8ON^+$ with the simulated fragments of $C_7H_7N$ with imine structure and of $C_7H_7ON$ with amide structure from Mass Frontier, respectively. Results show that the MS² spectra of $C_7H_8N^+$ and $C_7H_8ON^+$ matched well with the simulation results by Mass Frontier (Fig.2

d, f). These results demonstrate that the unstable $C_7H_9O_2N$ further decomposes into $C_7H_7N$ and $C_7H_7ON$. Previous theoretical study calculated that the reaction between NH₃ and $C_7$-SCI may produce the products $C_7H_7N$ and $C_7H_7ON$(Banu et al., 2018; Ma et al., 2018), which further supports our findings.

Combining multiple experimental evidence, we propose the following reaction mechanism between NH₃ and styrene-derived products and their role in SOA formation (Fig.2g). Styrene reacts with O₃ to form $C_7$-SCI, which then generates benzoic acid

and oligomerization to form SOA. However, the addition of NH₃ leads to a competitive reaction between both NH₃ and H₂O with $C_7$-SCI, forming an unstable peroxide amine $C_7H_9O_2N$, which rapidly further produces more stable imine $C_7H_7N$ and amide $C_7H_7ON$. Furthermore, due to the presence of peroxide bonds and nitrogen, toxicity calculations show that the toxicity of $C_7H_9O_2N$, $C_7H_7N$, and $C_7H_7ON$ (High, class III) is significantly higher than that of benzoic acid (Low, class I) based on Cramer classification(Cramer et al., 1976).

The reaction pathway between NH₃ and SCI identified in both isoprene and styrene systems indicates a general mechanism by which NH₃ affects SOA molecular composition across in different olefin VOCs, highlighting the widespread impact of NH₃ on aerosol chemistry, independent of the type of olefins. NH₃ entering aerosols through reaction results in the generation of NOCs (e.g., amines, imines), which changes aerosol composition and potentially enhances light absorption and toxicity.(Updyke et al., 2012) NH₃ reduces SOA yields but increases NOC diversity. In recent years, NH₃ emissions have

increased globally, driven by agricultural and industrial activities(Fu et al., 2017; Kuttippurath et al., 2020; Liu et al., 2018; Meng et al., 2020). Our study suggests that increasing NH₃ levels may suppress SOA from isoprene and styrene, and affect regional aerosol budgets. Further research is needed to determine whether it has an impact on other olefins. Current models ignore the role of NH₃ in SOA chemistry, and may overestimate the formation of SOA in NH₃-rich environments. Integrating the novel NOC formation pathway from NH₃ and SCI into the current model framework is crucial for improving climate and

health predictions of aerosols.

### Financial support

This research has been supported by the National Natural Science Foundation of China (42461160326, 42477492, 42175125), the Strategic Priority Research Program (B) of the Chinese Academy of Sciences (XDB0760200).



**Data availability**

The data that support the results can be found in the appendix of the supplementary material.

**Author contribution**

XL conducted experiments, data analysis, and drew graphs. LJ designs research, analyses data, and writes. YX designs and modifies papers. All the authors participated in writing the paper.

**Competing interests**

The contact author has declared that none of the authors has any competing interests.

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
