# Peer review of "NH3 Converts Criegee Intermediates to Nitrogenous Organics"

_EGUsphere, 2025_

## Author Comment (AC1)

**Response to Reviewer 1**

We greatly appreciate the time and effort the reviewer spent reviewing our manuscript. The comments are thoughtful and helpful in improving the quality of our paper. Below, we make a point-by-point response to these comments. The response to the reviewer is structured in the following sequence: (1) Comments from the reviewer in black color; (2) Our response in blue color; (3) Our changes in the revised manuscript in red color.

The authors present a study of styrene ozonolysis in the presence of ammonia, and report the suppression of SOA formation in the presence of ammonia and observations of species that indicate reactions of stabilised Criegee intermediates with ammonia.

The authors present some interesting results, but the manuscript is generally lacking in detail. At present the manuscript does little more than present observations. Real-time measurements are referred to, which indicate the potential to determine reaction kinetics, but none are reported. It would be beneficial to at least report timescales/kinetics for production of the species observed and, ideally, estimated yields that could be used in atmospheric models. The manuscript should be significantly improved prior to publication to include further details of the experiments and more comprehensive description of the analysis and modelling of results.

We thank the reviewer for providing valuable suggestions to enhance the quality of our paper. Based on the reviewer's valuable suggestions, we have provided more experimental details, conditions, and kinetic results in our revised manuscript. Specifically, we have moved the experimental details from the *supplementary material* to the *Materials and methods* section of the manuscript. In addition, we have provided point-to-point responses to the corresponding questions in the *Other Comments*. Furthermore, we provided a more detailed description of the analysis and modeling of the results, particularly in the determination of the reaction kinetics. The specific modifications are as follows.

Pages 9, Lines 177-196: Accurate quantification of C7H9O2N and its degradation

products typically requires the use of standard gases to establish a calibration coefficient between mass spectrometry signal abundance and actual concentration. However, due to the current unavailability of standard materials for C7H9O2N and its products, direct quantification is challenging. Nevertheless, a previous study (Ma et al., 2018) estimated the rate constant for the reaction of C7-SCI with NH3 forming C7H9O2N (1.65×10-15 cm3 molecule-1 s-1) via quantum chemical calculations. Based on this rate constant, we added the corresponding reaction into the MCM mechanism. Under Exp.10 experimental conditions, the simulated maximum concentration of C7H9O2N after 50 minutes of reaction was 28 ppb. Since the decomposition of C7H9O2N was not considered in the simulation, this concentration actually represents the total concentration of C7H9O2N and its two decomposition products. To further distinguish the specific concentrations of C7H9O2N and its two decomposition products, it needs to determine their decomposition rate constants. Fortunately, using online GAIS-Orbitrap MS monitoring data on abundance-time evolution, we can obtain the relative proportions among the three species: C7H9O2N (m/z 140): C7H7N (m/z 106): C7H7ON (m/z 122). Based on this ratio, we introduced two decomposition reactions into the MCM mechanism and adjusted their rate constants so that the simulated concentration ratios matched the experimentally observed values. The corresponding concentrations of  $C_7H_9O_2N$  (m/z 140),  $C_7H_7N$  (m/z 106) and  $C_7H_7ON$  (m/z 122) at the  $50^{th}$  minute were determined to be 23.8 ppb, 1.6 ppb and 2.7 ppb in Exp. 10, with a deviation of  $\pm 17\%$ . This allowed us to derive the two decomposition rate constants as  $(3.0\pm0.4)\times10^{-1}$  $^{5}$  s-1 and  $(5.1\pm0.6)\times10^{-5}$  s-1. To date, only Banu et al. (2018) have reported theoretical values for the two decomposition rate constants of C7H9O2N, which are 7.02×10-16 s-1 and 1.22×10-13 s-1, respectively. It shows that the experimentally derived decomposition rate constants are approximately eight orders of magnitude higher than the theoretical values, indicating that C7H9O2N is a highly unstable compound. Then, the maximum yields of C7H9O2N, C7H7N and C7H7ON can be determined to be 8.1%, 3.0%, and 5.1% in styrene-O3 system under conditions of Exp.10, respectively.

Other comments are listed below:

Line 14: Criegee intermediates should be more correctly referred to as zwitterions than radicals.

According to the reviewer's advice, we have revised "radicals" to "zwitterions" on Page 1, Line 14:

NH3 efficiently scavenges stable Criegee intermediates (SCI) - critical zwitterions in organic aerosol formation.

**Line 17: What is the expected atmospheric lifetime of the species C7H9O2N?**

The C7H9O2N molecule contains an unstable peroxide bond (-O-O-), which makes it highly reactive. Our experimental observations have confirmed its rapid decomposition into C7H7N and C7H7ON. In the actual atmosphere, in addition to self-decomposition, C7H9O2N may also react with OH radicals and undergo photolysis. After considering these 3 pathways, the atmospheric lifetime of C7H9O2N was estimated to be 2.1 hours. According to the reviewer's comments, we have provided a detailed analysis and explanation on Lines 197-205 of Pages 9-10.

To quantify the expected atmospheric lifetime of  $C_7H_9O_2N$ , we have considered 3 primary removal pathways: (1)Reaction with OH radicals, the reaction rate constant between  $C_7H_9O_2N$  and OH was estimated to be  $4.77\times10^{-11}$  cm3 molecule-1 s-1 using a tool of AOPWIN (Atmospheric Oxidation Program for Microsoft Windows) in EPI (Estimation Program Interface). Using an average OH radical concentration of  $1.0\times10^6$  molecules cm-3, the atmospheric lifetime of  $\tau_{OH} = 5.8$  hours; (2) Photolysis: Based on the general photolysis rates of peroxides  $1.3\times10^{-6}$  s-1 (Roehl et al., 2007), the photolytic lifetime  $\tau_{hv} = 214$  hours; (3) Thermal decomposition: Based on our results, the decomposition rate of  $C_7H_9O_2N$  is  $8.1\times10^{-5}$  s-1, and its self-decomposition lifetime  $\tau_{decomp} = 3.4$  hours. The total atmospheric lifetime was calculated to be 2.1 hours based on  $1/\tau = 1/\tau_{OH} + 1/\tau_{hv} + 1/\tau_{decom}$ . This suggests that  $C_7H_9O_2N$  predominantly exists in the atmosphere as its more stable transformation products, namely the imine  $C_7H_7N$  and the amide  $C_7H_7ON$ .

Line 35: What are typical emission rates or atmospheric concentrations of styrene in urban/industrial regions? How significant is the atmospheric loss of styrene to reaction with ozone compared to the reaction with OH?

The typical atmospheric concentration of styrene varies between urban and industrial areas from 0.06 to 45 ppb (Okada et al., 2012; Cho et al., 2014; Sun et al., 2016; Sheng et al., 2018). Ozone oxidation is an important atmospheric sink for styrene. Based on MCM mechanism, the rate constant ( $k_{OH}$ ) for the reaction of styrene with OH is  $5.8 \times 10^{-11}$  cm3molecule-1s-1 at 298 K, and the rate constant ( $k_{O3}$ ) for the reaction with O3 is  $1.7 \times 10^{-17}$  cm3molecule-1s-1 at 298 K. Under typical atmospheric conditions ([OH]  $\sim 1.0 \times 10^6$  molecule cm-3, [O3]  $\sim 50$  ppb= $1.3 \times 10^{12}$  molecule cm-3), the estimated loss ratio for the reaction of styrene with OH and O3 is about 2.6:1. This indicates that about 30% of styrene will be consumed by O3 in atmospheric conditions. According to the reviewer's comments, we have added the following content to the *Introduction* section.

Page 2, Lines 41-44: The typical atmospheric concentration of styrene varies between urban and industrial areas from 0.06 to 45 ppb (Okada et al., 2012; Cho et al., 2014; Sun et al., 2016; Sheng et al., 2018). Under typical atmospheric conditions, about 30% of styrene may be consumed by O3, thus ozone oxidation is an important sink for styrene, especially in areas with high O3 pollution.

Line 37: It would help to give the structures of the Criegee intermediates (and for other species discussed in the manuscript).

Based on the reviewer's suggestion. We have added the structures of Criegee intermediates and several other key species discussed in the manuscript on Page 1, Line 15 in the *supplementary material*:

Styrene
$$C_7$$
-SCI Benzoic acid

 $O_{NH_2}$
 $C_7H_9O_2N$
 $C_7H_7N$
 $C_7H_7ON$

Figure S1. The structures of the key species

**Line 52: Define FEP.**

We have defined FEP in the experiments section on Page 3, Line 60:

The chamber experiments were conducted in Fluorinated Ethylene Propylene (FEP, 200A, DuPont) reactors under dark conditions, with background air supplied by purified zero air.

Line 54: What were the typical concentrations/concentration ranges used in the experiments?

The reactants and their concentration ranges used in the exp.1-5 are: styrene (0.34~0.36 ppm), O3 (1 ppm), and NH3 (0~0.8 ppm), respectively. The concentration ranges used in the exp.6-10 are: styrene (0.4~0.7 ppm), O3 (2 ppm), and NH3 (0~10 ppm), respectively. The concentration used in the exp.11 is styrene (3 ppm), O3 (10 ppm), and NH3 (0.8 ppm), respectively. We have listed the initial concentrations used in each experiment in Table S1, and added the concentration ranges in the *Materials* and methods section.

Page 3, Lines 63-64: The reactants and their concentration ranges used in the experiment are styrene (0.3~3 ppm),  $O_3$  (1~10 ppm), and  $NH_3$  (0~10 ppm), respectively.

Line 75: It would help to provide more details of the experimental procedures and conditions in the main text. Which species were measured? What was the timescale for the measurements?

According to the reviewer's advice, we have moved some of the experimental details from the *supplementary material* to the *Materials and methods* of the main text in manuscript, and provided a detailed introduction to the species measured during the experiment and their corresponding timescales on Pages 3, Lines 60-80:

Experiments and Measurements: The chamber experiments were conducted in Fluorinated Ethylene Propylene (FEP, 200A, DuPont) reactors under dark conditions, with background air supplied by purified zero air. Styrene was injected into the reactor with zero air using a glass microsyringe, O3 was produced by an ozone generator with pure O2, and NH3 was directly injected into the reactor. The reactants and their concentration ranges used in the experiment are styrene (0.3~3 ppm), O3 (1~10 ppm), and NH3 (0~10 ppm), respectively. Because ozonolysis of styrene can form OH radicals, n-Hexane was used as an OH radical scavenger (>100ppm with a removal efficiency >90%). Detailed experimental conditions are provided in Table S1.

To collect particles and determine the SOA yields, experiments 1-5 were conducted in a 1.2 m3 chamber. During these experiments, styrene was measured online using a proton transfer reaction-mass spectrometer (PTR-MS P1000-L-AI, Anhui Province Key Laboratory of Medical Physics and Technology) with a time resolution of 20 s in the gas phase. O3 was measured every 0.5 hours lasting for 5 minutes with an O3 analyzer (Model 49C, Thermo Scientific) with a time resolution of 10 s in the gas phase. The particle concentrations and size distributions were determined by a scanning mobility particle sizer (SMPS, Model 3936, DMA-3080, CPC-3776, TSI) with a time resolution of 5 minutes. The online measurements covered the entire experimental process (4~5h). Particles were collected on a 25 mm polytetrafluoroethylene (PTFE) membrane with a pore size of 0.45 μm at the 4th hour, and the sample flow rate was 6 L/min and lasted for 40 min. The collected particles were extracted with methanol for composition analysis in the particle phase, which were injected by a high-performance

liquid chromatography (HPLC, Thermo Scientific), ionized by a heated electrospray ionization source (ESI), and then the molecular composition was measured by a high-resolution Orbitrap mass spectrometer (Orbitrap MS, Q-Exactive, Thermo Scientific) with a resolution R= 70,000 at m/z 200. To determine the kinetics and mechanism of the reaction between C7-SCI and NH3, experiments 6-10 were performed with higher concentrations in a 150 L chamber. During these experiments, the products were online ionized by a gas aerosol in-situ ionization source (GAIS), and then measured by Orbitrap MS in the gas phase. The time resolution of GAIS-Orbitrap MS measurement is about 0.5 s, and all the experiments lasted about 1 h.

Line 92: How certain is the mechanism for benzoic acid formation? It would help to show a schematic of the mechanism. Is there any evidence for combined effects of ammonia and water? Studies of Criegee intermediate kinetics using photolytic precursors have demonstrated cooperative effects of water and ammonia on Criegee intermediate chemistry (e.g. Chao et al., J. Phys. Chem. A, 123, 1337-1342, 2019).

The formation mechanism of benzoic acid through reaction between C7-SCI and H2O has been reported by previous studies (Na et al., 2006; Banu et al., 2018). In this study, the chemical mechanism of styrene was taken from the MCM mechanism v3.3.1 (http://mcm.york.ac.uk/), which is built using published experimental data, theoretical studies, and evaluated kinetic data (Jenkin et al., 2003). According to the reviewer's advice, we have added a schematic of the mechanism in Figure 1 d.

Thank you for recommending the important study by Chao et al. (2019), which revealed the strong synergistic effect of NH3 and H2O on the reaction of C1-SCI. We have cited this work in the *Introduction* section. In the present study, our primary focus is on the reaction between C7-SCI and NH3. It should be noted that the Q-Exactive mass spectrometer used here can only detect ions with an m/z greater than 50. Since the reaction products of C1-SCI with H2O or NH3 have molecular weights below 50 Da, we were unable to detect any of these products. Consequently, no synergic enhancement effect between water and NH3 was observed under our experimental

conditions. According to the reviewer's suggestions, we have revised the following content.

Page 2, Lines 33-38: Quantum calculations suggest that NH3 may influence the SOA formation from styrene through reactions with stable Criegee intermediates (SCIs) (Ma et al., 2018; Banu et al., 2018), and NH3 and H2O have a synergic effect on the reaction of C1-Criegee intermediate (Chao et al., 2019a,b). The reaction rate between NH3 and C1-Criegee intermediate (CH2OO) has been determined by theoretical calculations (Jørgensen and Gross, 2009; Misiewicz et al., 2018) and experiments (Liu et al., 2018; Chao et al., 2019; Chhantyal-Pun et al., 2019). Our recent study has shown new laboratory evidence that NH3 can also react with isoprene-derived SCIs to form NOCs, thereby changing the chemical characteristics of SOA (Li et al., 2024).

Page 4, Lines 115-116: Since benzoic acid is mainly formed from the reaction of C7-SCI with H2O (Na et al., 2006; Banu et al., 2018), the presence of NH3 apparently competes with H2O for SCIs and inhibits the formation of benzoic acid (Fig.1d).

Page 6, Line 131:

Figure 1: SOA mass yields from styrene ozonolysis under different NH3 concentrations (a); Positive mode mass spectra of SOA from styrene ozonolysis systems with 0 ppm (blue) and 0.4 ppm NH3 (red) (b), several top ion peaks assigned to SCI-derived oligomer are marked in black; The mass spectra of benzoic acid from styrene ozonolysis systems with 0 ppm (blue) and 0.4 ppm NH3 (red) (c); Online observation of benzoic acid in the experiments with low concentration NH3 with normal humidity (Ex.8, blue) and high concentration NH3 with low humidity (Ex.10, red) (d).

Line 140: What is 'the general decomposition principle of peroxides'? References and

details are needed here. What is known about the stability of the species being discussed?

Dehydration and dehydroperoxidation are common decomposition pathways for peroxides (Smith and March, 2020). Due to the high reactivity of peroxide bonds, the peroxide amine C7H9O2N is expected to be highly unstable and easily decomposed by removing one H2O2 or H2O (Smith and March, 2020; Banu et al., 2018; Ma et al., 2018).

Regarding the stability of species  $C_7H_9O_2N$ , our online mass spectrometry experimental evidence shows that the appearance of the  $C_7H_9O_2N$  signal is accompanied by an increase in the signals of its decomposition products ( $C_7H_7N$  and  $C_7H_7ON$ ), demonstrating the instability of  $C_7H_9O_2N$  on the experimental time scale. In addition, we also discussed the expected atmospheric lifetime of  $C_7H_9O_2N$ , and the result obtained was only 2.1 hours, further proving the instability of  $C_7H_9O_2N$ .

According to the reviewer's advice, we have added some sentences on Page 9, Lines 168-170:

Due to the high reactivity of peroxide bonds, the peroxide amine  $C_7H_9O_2N$  is expected to be highly unstable and easily decomposed by removing one  $H_2O_2$  or  $H_2O$  (Smith and March, 2020), and may further decompose into imines and amides based on theoretical calculation (Banu et al., 2018; Ma et al., 2018).

**References:**

Banu, T., Sen, K., and Das, A. K.: Atmospheric Fate of Criegee Intermediate Formed During Ozonolysis of Styrene in the Presence of H2O and NH3: The Crucial Role of Stereochemistry, J. Phys. Chem. A, 122, 8377–8389, https://doi.org/10.1021/acs.jpca.8b06835, 2018.

Chao, W., Yin, C., Takahashi, K., and Lin, J. J.-M.: Effects of water vapor on the reaction of CH2OO with NH3, Phys. Chem. Chem. Phys., 21, 22589–22597, https://doi.org/10.1039/C9CP04682H, 2019a.

Chao, W., Yin, C., Takahashi, K., and Lin, J. J.-M.: Hydrogen-Bonding Mediated Reactions of Criegee Intermediates in the Gas Phase: Competition between

Bimolecular and Termolecular Reactions and the Catalytic Role of Water, J. Phys. Chem. A, 123, 8336–8348, https://doi.org/10.1021/acs.jpca.9b07117, 2019b.

Chen, Y., Zhou, X., Liu, Y., Jin, Y., Dong, W., and Yang, X.: Kinetics of the simplest criegee intermediate CH2OO reacting with CF3CF=CF2, Chinese Journal of Chemical Physics, 33, 234–238, https://doi.org/10.1063/1674-0068/cjcp2002025, 2020.

Chhantyal-Pun, R., Shannon, R. J., Tew, D. P., Caravan, R. L., Duchi, M., Wong, C., Ingham, A., Feldman, C., McGillen, M. R., Khan, M. A. H., Antonov, I. O., Rotavera, B., Ramasesha, K., Osborn, D. L., Taatjes, C. A., Percival, C. J., Shallcross, D. E., and Orr-Ewing, A. J.: Experimental and computational studies of Criegee intermediate reactions with NH3 and CH3 NH2, Phys. Chem. Chem. Phys., 21, 14042–14052, https://doi.org/10.1039/C8CP06810K, 2019.

Cho, J., Roueintan, M., and Li, Z.: Kinetic and dynamic investigations of OH reaction with styrene, J. Phys. Chem. A, 118, 9460–9470, https://doi.org/10.1021/jp501380j, 2014.

Jenkin, M. E., Saunders, S. M., Wagner, V., and Pilling, M. J.: Protocol for the development of the Master Chemical Mechanism, MCM v3 (Part B): tropospheric degradation of aromatic volatile organic compounds, Part B, 2003.

Jørgensen, S. and Gross, A.: Theoretical Investigation of the Reaction between Carbonyl Oxides and Ammonia, J. Phys. Chem. A, 113, 10284–10290, https://doi.org/10.1021/jp905343u, 2009.

Liu, Y., Yin, C., Smith, M. C., Liu, S., Chen, M., Zhou, X., Xiao, C., Dai, D., Lin, J. J.-M., Takahashi, K., Dong, W., and Yang, X.: Kinetics of the reaction of the simplest Criegee intermediate with ammonia: a combination of experiment and theory, Phys. Chem. Chem. Phys., 20, 29669–29676, https://doi.org/10.1039/C8CP05920A, 2018.

Ma, Q., Lin, X., Yang, C., Long, B., Gai, Y., and Zhang, W.: The influences of ammonia on aerosol formation in the ozonolysis of styrene: roles of Criegee intermediate reactions, R. Soc. open sci., 5, 172171, https://doi.org/10.1098/rsos.172171, 2018.

Misiewicz, J. P., Elliott, S. N., Moore, K. B., and Schaefer, H. F.: Re-examining ammonia addition to the Criegee intermediate: converging to chemical accuracy, Phys.

Chem. Chem. Phys., 20, 7479–7491, https://doi.org/10.1039/C7CP08582F, 2018.

Na, K., Song, C., and Cockeriii, D.: Formation of secondary organic aerosol from the reaction of styrene with ozone in the presence and absence of ammonia and water, Atmos. Environ., 40, 1889–1900, https://doi.org/10.1016/j.atmosenv.2005.10.063, 2006.

Okada, Y., Nakagoshi, A., Tsurukawa, M., Matsumura, C., Eiho, J., and Nakano, T.: Environmental risk assessment and concentration trend of atmospheric volatile organic compounds in Hyogo Prefecture, Japan, Environ Sci Pollut Res, 19, 201–213, https://doi.org/10.1007/s11356-011-0550-0, 2012.

Roehl, C. M., Marka, Z., Fry, J. L., and Wennberg, P. O.: Near-UV photolysis cross sections of CH3OOH and HOCH2OOH determined via action spectroscopy, Atmos. Chem. Phys., 2007.

Sheng, J., Zhao, D., Ding, D., Li, X., Huang, M., Gao, Y., Quan, J., and Zhang, Q.: Characterizing the level, photochemical reactivity, emission, and source contribution of the volatile organic compounds based on PTR-TOF-MS during winter haze period in beijing, China, Atmospheric Research, 212, 54–63, https://doi.org/10.1016/j.atmosres.2018.05.005, 2018.

Smith, M. and March, J.: March's advanced organic chemistry: reactions, mechanisms, and structure, Eighth edition., John Wiley & Sons, Inc., Hoboken, New Jersey, 2020.

Sofen, E. D., Bowdalo, D., Evans, M. J., Apadula, F., Bonasoni, P., Cupeiro, M., Ellul, R., Galbally, I. E., Girgzdiene, R., Luppo, S., Mimouni, M., Nahas, A. C., Saliba, M., and Tørseth, K.: Gridded global surface ozone metrics for atmospheric chemistry model evaluation, 2016.

Sun, J., Wu, F., Hu, B., Tang, G., Zhang, J., and Wang, Y.: VOC characteristics, emissions and contributions to SOA formation during hazy episodes, Atmos. Environ., 141, 560–570, https://doi.org/10.1016/j.atmosenv.2016.06.060, 2016.

---

## Author Comment (AC2)

**Response to Reviewer 2**

We greatly appreciate the time and effort the reviewer spent reviewing our manuscript. The comments are thoughtful and helpful in improving the quality of our paper. Below, we make a point-by-point response to these comments. The response to the reviewer is structured in the following sequence: (1) Comments from the reviewer in black color; (2) Our response in blue color; (3) Our changes in the revised manuscript in red color.

Li et al. have utilized Orbitrap mass spectrometry to examine the products from the ozonolysis of styrene in the presence vs. absence of ammonia. In the presence of ammonia, the authors report the formation of a peroxide amine, which they attribute to the reaction of ammonia with the styrene-derived 7-carbon Criegee intermediate. They observe a reduction in other products that they attribute to Criegee intermediate-driven SOA formation, such as benzoic acid. This study provides new mechanistic insights on the potential contribution for ozonolysis driven intermediates to contribute to the formation of tropospheric SOA, but requires significant edits for clarity and accuracy. Comments and recommended edits follow:

We thank the reviewer for providing valuable suggestions to enhance the quality of our paper. Following the reviewer's suggestion, we have added sufficient experimental and simulation details in the *Materials and methods* section to enhance the clarity of the manuscript, and have added contents of calculation and analysis in the *Results and discussion* section to enhance the accuracy of the manuscript.

- Line 33: There have been several studies published on the reactions of Criegee intermediates with ammonia and amines that were conducted prior to the previous study by these authors which should be cited:
- S. Jørgensen, A. Gross, Theoretical Investigation of the Reaction between Carbonyl Oxides and Ammonia, JPCA (2009).

Liu, C. Yin, M. C. Smith, S. Liu, M. Chen, X. Zhou, C. Xiao, D. Dai, J. J.-M. Lin, K. Takahashi, W. Dong, X. Yang, Kinetics of the reaction of the simplest Criegee intermediate with ammonia: a combination of experiment and theory, PCCP (2018).

J. P. Misiewicz, S. N. Elliott, K. B. Moore, H. F. Schaefer, Re-examining ammonia addition to the Criegee intermediate: converging to chemical accuracy, PCCP (2018).

W. Chao, C. Yin, K. Takahashi, J. J. M. Lin, Effects of water vapor on the reaction of CH2OO with NH3, PCCP (2019).

R. Chhantyal-Pun, R. J. Shannon, D. P. Tew, R. L. Caravan, M. Duchi, C. Wong, A. Ingham, C. Feldman, M. R. McGillen, M. A. H. Khan, I. O. Antonov, B. Rotavera, K. Ramasesha, D. L. Osborn, C. A. Taatjes, C. J. Percival, D. E. Shallcross, A. J. Orr-Ewing, Experimental and computational studies of Criegee intermediate reactions with NH3 and CH3NH2. PCCP (2019).

Thanks for the reviewer. These references are very helpful for our manuscript. According to the reviewer's advice, we have added these contents on Page 2, Lines 33-38:

Quantum calculations suggest that NH3 may influence the SOA formation from styrene through reactions with stable Criegee intermediates (SCIs) (Ma et al., 2018; Banu et al., 2018). The reaction rate between NH3 and C1-Criegee intermediate (CH2OO) has been determined by theoretical calculations (Jørgensen and Gross, 2009; Misiewicz et al., 2018) and experiments (Liu et al., 2018; Chao et al., 2019a, b; Chhantyal-Pun et al., 2019). Our recent study has shown new laboratory evidence that NH3 can also react with isoprene-derived SCIs to form NOCs, thereby changing the chemical characteristics of SOA (Li et al., 2024).

Line 53 (and line 14 of the SM): define FEP.

We have defined FEP in the experiments section. Page 3, Line 60:

The chamber experiments were conducted in Fluorinated Ethylene Propylene (FEP, 200A, DuPont) reactors under dark conditions, with background air supplied by purified zero air.

Line 55: Please provide concentration ranges of styrene, O3, and NH3 used.

The reactants and their concentration ranges used in the exp.1-5 are: styrene (0.34~0.36 ppm), O3 (1 ppm), and NH3 (0~0.8 ppm), respectively. The concentration ranges used in the exp.6-10 are: styrene (0.4~0.7 ppm), O3 (2 ppm), and NH3 (0~10 ppm), respectively. The concentration used in the exp.11 is styrene (3 ppm), O3 (10 ppm), and NH3 (0.8 ppm), respectively. We have listed the initial concentrations used in each experiment in Table S1, and added the concentration ranges in the *Materials* and methods section on Page 3, Lines 63-64:

The reactants and their concentration ranges used in the experiment are styrene (0.3~3 ppm), O3 (1~10 ppm), and NH3 (0~10 ppm), respectively.

Line 72 (and SM Line 46): Which unimolecular and bimolecular reactions of the styrene-derived Criegee intermediates were included in the mechanism aside from the reaction with NH3?

In addition to the reaction with NH3 we have added, this mechanism also includes the pre-existing unimolecular reactions of styrene-derived Criegee intermediates to generate benzoic acid, benzaldehyde, H2O2 and HCOOH. And the pre-existing bimolecular reactions of styrene-derived Criegee intermediates with CO, NO, NO2 and SO2. Based on the reviewer's suggestions, we have added these contents to the manuscript and the *supplementary materials*.

Page 4, Lines 91-94: Gas-phase reactions were simulated using the Master Chemical Mechanism (MCM v3.3.1, website: https://mcm.york.ac.uk/MCM). To evaluate the influence of NH3-SCI reactions, we added four reactions to the MCM mechanism, including those between NH3 and C1-/C7-SCIs (CH2OO/PHCHOO) and the subsequent decomposition of C7H9O2N into C7H7N and C7H7ON.

Page 3-4, Lines 50-71 in *supplementary materials*: Gas-phase reactions were simulated using the Master Chemical Mechanism (MCM v3.3.1, website: https://mcm.york.ac.uk/MCM), the reaction between NH3 and C1-/C7-SCIs

(CH2OO/PHCHOO) and the subsequent decomposition of C7H9O2N into C7H7N and C7H7ON was introduced into MCM for simulation:

CH2OO+NH3= CH2OONH3

PHCHOO+NH3=PHCHOONH3

PHCHOONH3= PHCHNH

PHCHOONH3= PHCONH2

In addition to the reaction with NH3, MCM mechanism also includes the following unimolecular and bimolecular reactions of the styrene-derived Criegee intermediates:

PHCHOO+SO2=BENZAL+SO3

CO+PHCHOO=BENZAL

NO+PHCHOO=BENZAL+NO2

NO2+PHCHOO=BENZAL+NO3

PHCHOO=BENZAL+H2O2

PHCHOO=PHCOOH

CH2OO=HCOOH

CH2OO=H2O2+HCHO

CH2OO+CO=HCHO

CH2OO+NO=HCHO+NO2

CH2OO+NO2=HCHO+NO3

CH2OO+SO2=HCHO+SO3

Line 78: Do you use SOA yield to refer to SOA total mass, or number of particles?

The SOA yield mentioned refers to SOA mass yield, calculated based on the mass of reacted styrene and the total mass concentration of SOA. Based on the reviewer's suggestion, we have clarified SOA yields to SOA mass yields in the text and Figure 1 a.

Page 4, Lines 99-100: As NH3 concentrations increased, SOA mass yields

decreased significantly from (4.9±0.3)% (0 ppm NH3) to (1.0±0.1)% (0.8 ppm NH3), showing an obvious inhibitory effect (Fig.1a).

Line 78: What are the uncertainties on your reported 4.9% yield of SOA?

According to the reviewer's suggestion, we have included the instrumental uncertainties in the manuscript. The uncertainty of the PTR-MS (for styrene measurement) is  $\pm 6\%$ , and that of the SMPS (for aerosol mass concentration) is  $\pm 2\%$ . Based on the theory of error propagation, the relative uncertainty of the SOA yield is calculated to be  $\pm 6.32\%$ . The corresponding absolute uncertainty for the reported 4.9% yield is therefore  $\pm 0.3\%$ . The manuscript has been revised accordingly at the relevant positions on Page 4, Lines 99-100:

As NH3 concentrations increased, SOA mass yields decreased significantly from  $(4.9\pm0.3)\%$  (0 ppm NH3) to  $(1.0\pm0.1)\%$  (0.8 ppm NH3), showing an obvious inhibitory effect (Fig.1a).

Line 80: Provide the ranges of reported SOA yields for the studies cited to demonstrate that these are in line with the current work.

According to the reviewer's suggestion, we have added the ranges of SOA yields from the cited studies on Page 4, Lines 101-102:

The observed yields with 0 ppm NH3 are within the range of those previously reported for styrene ozonolysis under no NH3 conditions (2.7%~6.5%) (Bracco et al., 2019; Díaz-de-Mera et al., 2017; Yu et al., 2022, 2024b), which demonstrates the reasonability of our experiments.

Line 83: Suggest rephrasing "breaks up" with "competes with" or similar for clarity.

According to the reviewer's suggestion, we have replaced the "breaks up" on Page 4, Lines 104-106:

In the styrene-O3 reaction system, SOA is primarily derived from SCI-related products. As the concentration of NH3 increases, the SOA concentration decreases linearly. This indicates that the observed reduction of SOA is attributed to the competitive consumption of SCI by NH3.

Line 89: Do you see any evidence in the mass spectra for the sequential insertion of the 1 and/or 7 carbon Criegee intermediates into the initial peroxide amine reaction product (e.g., oligomerization).

We have carefully reexamined the MS data of SOA from styrene-O3-NH3 systems, and did not find any products that formed by insertion of C1 or C7 Criegee intermediates into the initial peroxide amine.

Line 92: Please provide a reference for a study that demonstrates the formation of benzoic acid from the C7 Criegee intermediate with water.

According to the reviewer's suggestion, we have added corresponding references.

Page 4, Lines 115-116: Since benzoic acid is mainly formed from the reaction of C7-SCI with H2O (Na et al., 2006; Banu et al., 2018), the presence of NH3 apparently competes with H2O for SCIs and inhibits the formation of benzoic acid (Fig.1d).

Line 95: Do you see any evidence for the water-mediated enhancement of the Criegee intermediate reaction with ammonia in your experiments, as reported by the Lin group? (I imagine that the MCM mechanism you are using in your analysis doesn't include this process):

W. Chao, C. Yin, Y.-L. Li, K. Takahashi, J. J.-M. Lin, Synergy of Water and Ammonia Hydrogen Bonding in a Gas-Phase Reaction, JPCA (2019)

W. Chao, C. Yin, K. Takahashi, J. J.-M. Lin, Effects of water vapor on the reaction of CH2OO with NH3, PCCP (2019)

We thank the reviewer for pointing out these important findings from Chao et al.

2019). This research reveals that NH3 and H2O have a synergic effect on the reaction of CH2OO. We have cited the work of Chao et al. in the introduction section. In the present study, our primary focus is on the reaction between C7-SCI and NH3. It should be noted that the Q-Exactive mass spectrometer used here can only detect ions with an m/z greater than 50. Since the reaction products of C1-SCI with H2O or NH3 have molecular weights below 50 Da, we were unable to detect any of these products. Consequently, no synergic enhancement effect between H2O and NH3 was observed under our experimental conditions. Hence, the MCM mechanism did not include synergistic promotion effect in this study. According to the reviewer's suggestions, we have revised the following content on Page 2, Lines 33-39:

Quantum calculations suggest that NH3 may influence the SOA formation from styrene through reactions with stable Criegee intermediates (SCIs) (Ma et al., 2018; Banu et al., 2018), and NH3 and H2O have a synergic effect on the reaction of C1-Criegee intermediate (Chao et al., 2019a,b). The reaction rate between NH3 and C1-Criegee intermediate (CH2OO) has been determined by theoretical calculations (Jørgensen and Gross, 2009; Misiewicz et al., 2018) and experiments (Liu et al., 2018; Chao et al., 2019; Chhantyal-Pun et al., 2019). Our recent study has shown new laboratory evidence that NH3 can also react with isoprene-derived SCIs to form NOCs, thereby changing the chemical characteristics of SOA (Li et al., 2024).

Figure 1 (upper): please add error bars.

According to the reviewer's suggestion, we have added error bars in Figure 1 a.

Figure 1: SOA mass yields from styrene ozonolysis under different NH3 concentrations (a); Positive mode mass spectra of SOA from styrene ozonolysis systems with 0 ppm (blue) and 0.4 ppm NH3 (red) (b), several top ion peaks assigned to SCI-derived oligomer are marked in black; The mass spectra of benzoic acid from styrene ozonolysis systems with 0 ppm (blue) and 0.4 ppm NH3 (red) (c); Online observation of benzoic acid in the experiments with low concentration NH3 with normal humidity (Ex.8, blue) and high concentration NH3 with low humidity (Ex.10, red) (d).

Figure 1 (middle): Are there any mass peaks which do not have changes in intensity in the presence vs absence of ammonia?

Yes, there are some peaks whose intensity remain almost unchanged. These primarily correspond to some species not directly involved in the SCI or NH3 reaction pathways. For example, the intensity of benzaldehyde ( $C_7H_6O$ , m/z=107.049) are  $5.6\times10^6$  and  $6.4\times10^6$  in the presence vs absence of ammonia, respectively, whose difference is within the uncertainty of  $\pm15\%$  in MS measurement.

Figure 1(lower): Please can you explain the rationale for changing the concentration of both ammonia and water in the red vs. blue datasets, rather than keeping one of the co-reactant concentrations the same.

These experiments were designed to maximally reveal the potential of NH3 and H2O to compete for C7-SCI under extreme conditions. By contrasting the two very different conditions of low NH3/normal humidity vs. high NH3/extremely low humidity, the strong suppression on the formation of benzoic acid can be most clearly demonstrated when the concentration of NH3 is much higher than that of H2O. This is not to directly compare rates, but to qualitatively verify the existence of competition mechanisms. According to the reviewer's comment, we have revised the following sentence on Page 5, Line 117-119.

To maximize the potential of NH3 and H2O to compete for C7-SCI under extreme conditions, we further conducted two experiments with low NH3/normal humidity vs. high NH3/extremely low humidity. The strong suppression on the formation of benzoic acid can be most clearly demonstrated when the concentration of NH3 is much higher than that of H2O.

Line 112: Please cite the (aforementioned) papers prior to the 2024 Li et al. work where the reaction mechanism for the reaction of Criegee intermediates with ammonia has been deduced.

According to the reviewer's advice, we have added the citation and the contents on Page 7, Lines 138-140:

Referring to the reaction mechanism between C1-SCI and NH3 (Jørgensen and Gross, 2009; Misiewicz et al., 2018; Liu et al., 2018; Chao et al., 2019a,b; Chhantyal-Pun et al., 2019), and the reaction mechanism between C4-SCI and NH3 from isoprene (Li et al., 2024), C7-SCI should react with NH3 to produce a molecule C7H9O2N.

Line 127: Please provide details of the iodometry kinetics measurements in Section 2.

According to the reviewer's advice, we have moved the details of the iodometry kinetics measurements from the *supplementary material* to the *Materials and methods* of the manuscript on Page 3, Lines 81-87:

To detect peroxides in the sample, experiment 11 was conducted in the 1.2 m3 chamber. The collected sample was immediately extracted by 400  $\mu$ L acetonitrile (ACN) before being injected into HPLC-HRMS. Using ACN as extraction solvent to minimize other unwanted decomposition processes such as hydrolysis. Half of the liquid (180  $\mu$ L) from the combined extract mixed with 10  $\mu$ L acetic acid (600 mM in ACN) in a vial, followed by the addition of 10  $\mu$ L KI (99.5%, Sigma-Aldrich) (400 mM in H2O) to trigger the iodometry reaction; another 180  $\mu$ L aliquot was treated in a same way by adding 10  $\mu$ L acetic acid (600 mM in ACN) and 10  $\mu$ L H2O, instead of KI. These two SOA samples are designated as KI-treated and non-treated respectively, which were injected into HPLC-HRMS (Li et al., 2025).

Figure 2 (a-g): Changing between labelling the (detected) protonated m/z and the (actual) product m/z in these figures is confusing. Perhaps the labelling can be made clearer, for example in (g) by providing both the actual molecular mass, and the mass at which the molecule was detected.

According to the reviewer's advice, we have revised a clearer labeling by providing both the actual molecular mass, and the mass at which the molecule was

**detected in Figure 2.**

Figure 2: Time series of online observation (a) and the chromatograms of molecule C7H9O2N (MW 139, m/z 140.071) are shown as an inset with the initial non-KI-treated sample (black) and KI-treated sample (red). MS2 spectra of C7H9O2N (MW 139, m/z 140.071, blue) and C7H6O2 (MW 122, m/z 123.044, black) in positive modes (b). Time series of online observation of C7H7N and C7H7ON (c, e). The comparison of the ion peaks in the MS2 spectra of C7H8N+ and C7H8ON+ (black bars) with the major simulated product ions of C7H7N and C7H7ON (red bars) (d, f). The mechanism of NH3 effects on SOA from styrene ozonolysis in this study (g).

Figure 2(g): Oligomers in the context of Criegee intermediates typically mean products that result from the insertion of several Criegee intermediate molecules into a

single co-reactant molecule, via sequential reactions. I believe that you are here referring to products resulting from the self-reaction(s) of Criegee intermediates, and so this distinction should be clearly made.

Thanks for the reviewer's comments. The structure we drew in Figure 2(g) did not clearly represent the oligomers we originally intended to explain. According to the reviewer's advice, we have revised the structure in Figure 2(g).

Line 139: Peroxides have been detected in both laboratory studies and in the field in the gas phase. Why do you think that this peroxide amine (which you detect here) would not survive under atmospherically relevant conditions? Please provide references for your proposed decomposition mechanism for this peroxide under atmospherically relevant conditions.

The C7H9O2N molecule contains an unstable peroxide bond (-O-O-), which makes it highly reactive. Our experimental observations have confirmed its rapid decomposition into C7H7N and C7H7ON. As stated in our response to the first reviewer, based on the experimental measurements and theoretical calculations in the reference, we have determined the rate constants for the decomposition of C7H9O2N into C7H7N and C7H7ON are (3.0±0.4)×10-5 s-1 and (5.1±0.6)×10-5 s-1, respectively (Page 9, Lines 177-196). In the actual atmosphere, in addition to self-decomposition, C7H9O2N may also react with OH radicals and undergo photolysis. After considering these 3 consumption pathways, the atmospheric lifetime of C7H9O2N was estimated to be 2.1 hours. According to the reviewer's comments, we have provided a detailed analysis and explanation on Lines 199-210 of Page 10, and added references on Page 9, Line 168-172.

Pages 9-10, Lines 197-205: The C7H9O2N molecule contains an unstable peroxide bond (-O-O-), which makes it highly reactive and short-lived in the atmosphere (Smith and March, 2020). Our experimental observations have confirmed its rapid decomposition into more stable imines (C7H7N) and amides (C7H7ON). To quantify the expected atmospheric lifetime of C7H9O2N, we have considered 3 primary removal

pathways: (1) Reaction with OH radicals, the reaction rate constant between  $C_7H_9O_2N$  and OH was estimated to be  $4.77\times10^{-11}$  cm3 molecule-1 s-1 using a tool of AOPWIN (Atmospheric Oxidation Program for Microsoft Windows) in EPI (Estimation Program Interface). Using an average OH radical concentration of  $1.0\times10^6$  molecules cm-3, the atmospheric lifetime of  $\tau_{OH} = 5.8$  hours; (2) Photolysis: Based on the general photolysis rates of peroxides  $1.3\times10^{-6}$  s-1 (Roehl et al., 2007), the photolytic lifetime  $\tau_{hv} = 214$  hours; (3) Thermal decomposition: Based on our results, the decomposition rate of  $C_7H_9O_2N$  is  $8.1\times10^{-5}$  s-1, and its self-decomposition lifetime  $\tau_{decomp} = 3.4$  hours. The total atmospheric lifetime was calculated to be 2.1 hours based on  $1/\tau = 1/\tau_{OH} + 1/\tau_{hv} + 1/\tau_{decom}$ . This suggests that  $C_7H_9O_2N$  predominantly exists in the atmosphere as its more stable transformation products, namely the imine  $C_7H_7N$  and the amide  $C_7H_7ON$ .

Page 9, line 168-170: Due to the high reactivity of peroxide bonds, the peroxide amine C7H9O2N is expected to be highly unstable and easily decomposed by removing one H2O2 or H2O (Smith and March, 2020), and may further decompose into imines and amides based on theoretical calculation (Banu et al., 2018; Ma et al., 2018).

Lines 149-150: Please provide supporting references that the C7 Criegee intermediates definitively lead to the formation of SOA via benzoic acid and oligomers, or adjust your statement accordingly.

According to the reviewer's suggestions, we have added citations supporting the contribution of C7- Criegee intermediates to SOA through oligomerization, such as Yu et al., 2022 and Yu et al., 2025, and revised the following sentence on Page 10, Lines 207-208:

Styrene reacts with O3 to form C7-SCI, which then generates benzoic acid and forms SOA through oligomerization (Yu et al., 2022; Yu et al., 2025).

Line 151: You state that the peroxide amine will 'rapidly' decompose to an imine and amide, but you detect the peroxide amine in the present work. Given your experimental conditions, can you determine a lower limit of the lifetime of the peroxide

**amine?**

The C7H9O2N molecule contains an unstable peroxide bond (-O-O-), which makes it highly reactive. Our experimental observations have confirmed its rapid decomposition into C7H7N and C7H7ON. As stated in our response to the first reviewer, based on the experimental measurements and theoretical calculations in the reference, we have determined the rate constants for the decomposition of C7H9O2N into C7H7N and C7H7ON are  $(3.0\pm0.4)\times10^{-5}$  s-1 and  $(5.1\pm0.6)\times10^{-5}$  s-1, respectively (Page 9, Lines 177-196). In the actual atmosphere, in addition to self-decomposition, C7H9O2N may also react with OH radicals and undergo photolysis. After considering these 3 pathways, the atmospheric lifetime of C7H9O2N was estimated to be 2.1 hours. We have provided a detailed analysis and explanation on Lines 197-205 of Page 10.

Line 163: Does your analysis account for the nucleation of peroxide amines onto existing particles, or just for new particle formation?

We did not add any additional seeds to the experimental system. In our previous research (Yu et al., 2022), we have found that styrene ozonolysis can produce extremely low-volatility compounds responding to nucleation. The molecular weight of the species C7H9O2N in this study is too small, mainly entering the particle phase through gas-particle distribution.

General comment: it is unclear in the manuscript which species were detected in the gas or particle phase.

Styrene was measured online using a proton transfer reaction-mass spectrometer in the gas phase. O3 was measured with an O3 analyzer in the gas phase. The collected particles were extracted for composition analysis in the particle phase. The species C7H9O2N, C7H7N and C7H7ON were online ionized by a gas aerosol in-situ ionization source (GAIS), and then measured by Orbitrap MS in the gas phase. According to the reviewer's suggestions, we have revised the following sentence on Page 3, Lines 67-80:

During these experiments, styrene was measured online using a proton transfer reaction-mass spectrometer (PTR-MS P1000-L-AI, Anhui Province Key Laboratory of Medical Physics and Technology) with a time resolution of 20 s in the gas phase. O3 was measured every 0.5 hours lasting for 5 minutes with an O3 analyzer (Model 49C, Thermo Scientific) with a time resolution of 10 s in the gas phase. The particle concentrations and size distributions were determined by a scanning mobility particle sizer (SMPS, Model 3936, DMA-3080, CPC-3776, TSI) with a time resolution of 5 minutes. The online measurements covered the entire experimental process (4~5h). Particles were collected on a 25 mm polytetrafluoroethylene (PTFE) membrane with a pore size of 0.45 µm at the 4th hour, and the sample flow rate was 6 L/min and lasted for 40 min. The collected particles were extracted with methanol for composition analysis in the particle phase, which were injected by a high-performance liquid chromatography (HPLC, Thermo Scientific), ionized by a heated electrospray ionization source (ESI), and then the molecular composition was measured by a highresolution Orbitrap mass spectrometer (Orbitrap MS, Q-Exactive, Thermo Scientific) with a resolution R = 70,000 at m/z 200. To determine the kinetics and mechanism of the reaction between C7-SCI and NH3, experiments 6-10 were performed with higher concentrations in a 150 L chamber. During these experiments, the products were online ionized by a gas aerosol in-situ ionization source (GAIS), and then measured by Orbitrap MS in the gas phase. The time resolution of GAIS-Orbitrap MS measurement is about 0.5 s, and all the experiments lasted about 1 h.

SM Line 22: Please provide concentration of n-hexane used for OH scavenging.

According to the reviewer's suggestion, we have added the sentence on Page 3, Lines 64-65:

Because ozonolysis of styrene can form OH radicals, n-Hexane was used as an OH radical scavenger (>100ppm with a removal efficiency >90%). Detailed experimental conditions are provided in Table S1.

**References:**

Banu, T., Sen, K., and Das, A. K.: Atmospheric Fate of Criegee Intermediate Formed During Ozonolysis of Styrene in the Presence of H2O and NH3: The Crucial Role of Stereochemistry, J. Phys. Chem. A, 122, 8377–8389, https://doi.org/10.1021/acs.jpca.8b06835, 2018.

Bracco, L. L. B., Tucceri, M. E., Escalona, A., Díaz-de-Mera, Y., Aranda, A., Rodríguez, A. M., and Rodríguez, D.: New particle formation from the reactions of ozone with indene and styrene, Phys. Chem. Chem. Phys., 21, 11214–11225, https://doi.org/10.1039/C9CP00912D, 2019.

Chao, W., Yin, C., Takahashi, K., and Lin, J. J.-M.: Effects of water vapor on the reaction of CH2OO with NH3, Phys. Chem. Chem. Phys., 21, 22589–22597, https://doi.org/10.1039/C9CP04682H, 2019a.

Chao, W., Yin, C., Takahashi, K., and Lin, J. J.-M.: Hydrogen-Bonding Mediated Reactions of Criegee Intermediates in the Gas Phase: Competition between Bimolecular and Termolecular Reactions and the Catalytic Role of Water, J. Phys. Chem. A, 123, 8336–8348, https://doi.org/10.1021/acs.jpca.9b07117, 2019b.

Chen, Y., Zhou, X., Liu, Y., Jin, Y., Dong, W., and Yang, X.: Kinetics of the simplest criegee intermediate CH2OO reacting with CF3CF=CF2, Chinese Journal of Chemical Physics, 33, 234–238, https://doi.org/10.1063/1674-0068/cjcp2002025, 2020.

Chhantyal-Pun, R., Shannon, R. J., Tew, D. P., Caravan, R. L., Duchi, M., Wong, C., Ingham, A., Feldman, C., McGillen, M. R., Khan, M. A. H., Antonov, I. O., Rotavera, B., Ramasesha, K., Osborn, D. L., Taatjes, C. A., Percival, C. J., Shallcross, D. E., and Orr-Ewing, A. J.: Experimental and computational studies of Criegee intermediate reactions with NH3 and CH3 NH2, Phys. Chem. Chem. Phys., 21, 14042–14052, https://doi.org/10.1039/C8CP06810K, 2019.

Díaz-de-Mera, Y., Aranda, A., Martínez, E., Rodríguez, A. A., Rodríguez, D., and Rodríguez, A.: Formation of secondary aerosols from the ozonolysis of styrene: Effect of SO2 and H2O, Atmos. Environ., 171, 25–31, https://doi.org/10.1016/j.atmosenv.2017.10.011, 2017.

Jørgensen, S. and Gross, A.: Theoretical Investigation of the Reaction between Carbonyl Oxides and Ammonia, J. Phys. Chem. A, 113, 10284–10290, https://doi.org/10.1021/jp905343u, 2009.

Li, K., Zheng, Z., Resch, J., Ma, J., Hansel, A., and Kalberer, M.: Molecular composition of organic peroxides in secondary organic aerosols revealed by peroxide-iodide reactivity, Environ. Sci. Technol., 59, 17126–17136, https://doi.org/10.1021/acs.est.5c03241, 2025.

Li, X., Jia, L., Xu, Y., and Pan, Y.: A novel reaction between ammonia and Criegee intermediates can form amines and suppress oligomers from isoprene, Science of The Total Environment, 956, 177389, https://doi.org/10.1016/j.scitotenv.2024.177389, 2024.

Liu, Y., Yin, C., Smith, M. C., Liu, S., Chen, M., Zhou, X., Xiao, C., Dai, D., Lin, J. J.-M., Takahashi, K., Dong, W., and Yang, X.: Kinetics of the reaction of the simplest Criegee intermediate with ammonia: a combination of experiment and theory, Phys. Chem. Chem. Phys., 20, 29669–29676, https://doi.org/10.1039/C8CP05920A, 2018.

Misiewicz, J. P., Elliott, S. N., Moore, K. B., and Schaefer, H. F.: Re-examining ammonia addition to the Criegee intermediate: converging to chemical accuracy, Phys. Chem. Chem. Phys., 20, 7479–7491, https://doi.org/10.1039/C7CP08582F, 2018.

Na, K., Song, C., and Cockeriii, D.: Formation of secondary organic aerosol from the reaction of styrene with ozone in the presence and absence of ammonia and water, Atmos. Environ., 40, 1889–1900, https://doi.org/10.1016/j.atmosenv.2005.10.063, 2006.

Tajuelo, M., Rodríguez, D., Baeza-Romero, M. T., Díaz-de-Mera, Y., Aranda, A., and Rodríguez, A.: Secondary organic aerosol formation from styrene photolysis and photooxidation with hydroxyl radicals, Chemosphere, 231, 276–286, https://doi.org/10.1016/j.chemosphere.2019.05.136, 2019.

Yu, S., Jia, L., Xu, Y., and Pan, Y.: Formation of extremely low-volatility organic compounds from styrene ozonolysis: Implication for nucleation, Chemosphere, 305, 135459, https://doi.org/10.1016/j.chemosphere.2022.135459, 2022.

Yu, S., Tong, S., Chen, M., Zhang, H., Xu, Y., Guo, Y., and Ge, M.: Characterization of key intermediates and products from the ozonolysis of styrene-like compounds, Environ. Sci. Technol., 59, 11666–11676, https://doi.org/10.1021/acs.est.5c00769, 2025.